# Influence of Osteopathic Manipulative Treatment on the Quality of Life and the Intensity of Lumbopelvic Pain in Pregnant Women in the Third Trimester: A Prospective Observational Study

**DOI:** 10.3390/healthcare11182538

**Published:** 2023-09-14

**Authors:** Maria Luisa Arruda Correia, Fernando Maia Peixoto Filho, Saint Clair Gomes Júnior

**Affiliations:** 1Department of Applied Research in Women’s Health, IFF/Fiocruz, Rio de Janeiro 22250-020, Brazil; 2Department of Fetal Medicine, IFF/Fiocruz, Rio de Janeiro 22250-020, Brazil; fernando.maia.peixoto.filho@uerj.br; 3Department of Clinical Research, A Researcher at IFF/Fiocruz, Rio de Janeiro 22250-020, Brazil; saintclair.junior@iff.fiocruz.br

**Keywords:** quality of life, pregnancy, health-related quality of life, osteopathic manipulative treatment, antenatal care, visual pain scale

## Abstract

During pregnancy, the various changes women undergo can affect their health status. Manual therapies are important aids because they do not use medication. This study aimed to evaluate the influence of osteopathic manipulative treatment on the intensity of lumbar and pelvic pain and changes in quality of life. This prospective study included women over 18 years old and between 27 and 41 weeks pregnant, and excluded women with fetal malformations, multiple fetuses, premature rupture of membranes, and in labor. Forty-six pregnant women were selected and divided into two groups of ≤3 and ≥4 visits. Statistically significant improvements were observed in the intensity of maximum low back pain (7.54 ± 1.47 vs. 3.815 ± 1.73, *p* ≤ 0.01) and minimum low back pain (5.67 ± 2.03 vs. 3.111 ± 1.67, *p* ≤ 0.01), maximum pelvic pain (6.54 ± 2.22 vs. 2.77 ± 1.64, *p* = 0.01), and minimum pelvic pain (5.615 ± 2.21 vs. 2.615 ± 1.66, *p* = 0.01). Both groups achieved improvements in quality of life indices, with the improvements achieved by the ≥4-visits group being statistically significant. Osteopathic treatment was effective in reducing the intensity of lumbar and pelvic pain and in improving the quality of life of pregnant women in the third trimester.

## 1. Introduction

Pregnancy is a state that causes several hormonal, physiological, and postural changes in the female body [1]. These changes can cause discomfort that affects the pregnant woman’s quality of life (QoL) and interfere with its physical, mental, and/or social aspects [2]. According to the WHO, a positive pregnancy experience includes physical and sociocultural normality and maintaining a healthy pregnancy for the mother and baby [3]. Within an individual-based view of health, looking after a pregnant woman’s QoL is as important as attention to the mother’s and baby’s mortality/morbidity rates [4].

Within a biopsychosocial view, it is essential to consider the intrinsic and extrinsic factors capable of altering the pain conditions and quality of life of pregnant women. External influences such as living conditions and environmental impacts should also be considered.

Intrinsic factors, such as cervical, lumbar, and pelvic pain, affect most pregnant women and directly influence their QoL. According to studies, pain affects 20% to 90% of pregnant women, and the intensity and location of the pain may even vary according to the country studied [5,6], in addition to widely recognized factors such as obesity, lifestyle, and work. In 20% of pregnant women, pain conditions can be severe enough to require therapeutic intervention [6]. However, pharmacological treatments available to pregnant women are limited, and the adoption of non-drug approaches, such as osteopathic manipulative treatment (OMT), becomes a feasible option.

OMT uses many manual techniques to improve physiological functions and support homeostasis altered by somatic dysfunction [7]. Studies already conducted on OMT in pregnant women have shown it to be safe and effective in controlling low back pain in pregnant women [8,9,10,11,12,13]. Physical and mental benefits of the use of OMT in pregnant women have also been observed [14].

Several validated instruments are used to measure the QoL of pregnant women, such as the SF-36, which, although not specific, is widely used in different populations, including pregnant women [2,4,15]. The eight health domains in the QoL SF-36 represent the health aspects that are most evaluated and affected by the diseases or therapies used [16].

However, few data relate variations in the QoL of pregnant women to the end of pregnancy, where issues related to low back and pelvic pain are more evident. Therefore, the objective of this research was to evaluate the intensity of lumbar and pelvic pain and changes in the QoL of pregnant women undergoing OMT.

## 2. Materials and Methods

### 2.1. Study Design

This prospective observational study assessed the influence of OMT on the QoL and the intensity of low back and pelvic pain in pregnant women in the third trimester of pregnancy. The survey was carried out between August 2021 and September 2022, during the COVID-19 pandemic in Brazil.

Pregnant women were screened at the prenatal outpatient clinic of Instituto Fernandes Figueira/Fiocruz (IFF), a tertiary health unit specialized in women’s, children’s, and adolescent health. The ethics committee of the IFF approved the research under the number 32216620.0.0000.5269, and all pregnant women signed a free consent form before participating in the study.

### 2.2. Inclusion and Exclusion Criteria

All pregnant women over 18 years of age from 27 weeks of gestation who were followed up by the prenatal service of the institution were considered eligible. Pregnant women with multiple pregnancies, malformation or congenital anomalies, premature rupture of the membrane, or in labor, and those with fewer than two QoL assessments were excluded.

Gestational age was determined by the date of the first-trimester ultrasound according to FIGO 2021 recommendations [17]. Hypertensive disorders of pregnancy were defined by the criteria of the American College of Obstetricians and Gynecologists [18]. BMI data followed the requirements of the Institute of Medicine (IOM, 2009) [19]. All pregnant women’s clinical, obstetric, and demographic data were taken from the patient’s medical records.

OMT consultations were linked to appointments with the obstetrics team at the institution’s prenatal clinic. They were carried out at the outpatient clinic, in a separate room, on the same day as the medical appointments. The time between consultations varied depending on the clinical condition of the pregnant woman and the fetus and the length of the pregnancy. In general, in pregnancies of usual risk, the return was one month (for pregnant women between 28 and 32 weeks of GA), then biweekly (for pregnant women between 32 and 36 weeks of GA), and then weekly in the last month of the gestation period (pregnant women with GA ranging from 36 to 40 weeks).

To enable enough time for completing the QoL SF-36, the patients’ entry into the study had a deadline of 36 weeks of gestational age (GA). 

### 2.3. Procedures

#### 2.3.1. Pandemic Procedures

As a result of the period of care occurring during the pandemic, the use of masks in hospital facilities became mandatory for staff and patients. When vaccines began to be made available through the public health system, proof of vaccination was also required to enter the hospital.

#### 2.3.2. OMT

All patients received the same protocol designed for this study, which included osteopathic techniques for balancing ligament tension (BLT) as well as myofascial, muscle energy, and cranial tension. High-velocity and low-amplitude (HVLA) techniques were excluded. OMT sessions ranged from 30 to 40 min and were performed by a single osteopath D.O. with 20 years of practice. The pregnant women were instructed to maintain standard obstetric and pharmacological treatments, and the OMT protocol with the techniques used is described in the Appendix A.

The OMT consultations took place following the hospital protocol, which recommended keeping windows open to improve the ventilation of the environment and minimize the possibility of contagion. Patients were treated dressed and placed on disposable sheets.

#### 2.3.3. QoL SF-36

The validated Brazilian version of the QoL SF-36 (Medical Outcomes Study 36—Item Short-Form Health Survey) was administered once during each visit to the prenatal outpatient clinic, consistently following the application of OMT. It is a self-administered, multidimensional instrument comprising 36 items, encompassed in 8 scales or components: functional capacity, physical aspects, pain, general health, vitality, social factors, emotional elements, and mental health. The score ranges from 0 to 100, where zero corresponds to the worst general health status and 100 to the best health status [19]. The Brazilian version of the QoL SF-36 was adapted using simple Portuguese, matching our cultural values. It was compared with the German, French, and Swedish versions, with cultural adaptations such as changing measurements from miles to kilometers and assessment activities such as functional ability. Inter and intra-observer reproducibility was observed. It can be seen in Appendix A.

#### 2.3.4. Visual Pain Scale (VAS)

The Visual Analogue Scale (VAS) is a unidimensional instrument used to assess pain intensity. It is a 10 cm line with pain scales ranging from 0 to 10, divided into mild (0 to 2), moderate (3 to 7), and severe (8 to 10) groups. The scale also uses visual resources such as drawings representing facial expressions [20]. In all consultations, the patient was asked about the presence of pain. If the patient answered positively, the scale was presented to the patient (Appendix A).

Low back pain was defined as pain located between the last posterior ribs down to the sacral promontory, and pelvic pain as pain located in the region of the pelvic girdle (pubis, perineum, groin, pelvis, and hip joints).

### 2.4. Statistical Analysis

Categorical variables were analyzed by the absolute and relative frequencies of occurrence and numeric variables by the mean and standard deviation. The stratify of the sample was decided after the initial analyses. The groups were defined considering the number of OMTs performed on the ≤3 and ≥4 sections.

The chi-square test was used to identify statistically significant differences in categorical variables and the Mann–Whitney test for numerical variables without normal distribution. The Wilcoxon test for paired samples was performed to evaluate differences in the range of the quality of life. All analyses were performed in SPSS v23, and *p*-value < 0.05 was adopted to characterize statistically significant differences.

## 3. Results

A total of sixty-five pregnant women were considered eligible, one of whom was excluded due to fetal malformation confirmed after birth. Due to the necessity of comparing de data of QoL, of the 64 pregnant women, 46 met the eligibility criterion of a minimum of two QoL SF-36 assessments. The pregnant women were divided into two groups according to their number of visits, i.e., up to ≤3 (22) and ≥4 visits (24). Figure 1 shows this study’s flowchart with the number of pregnant women. Data analysis on pain intensity was linked to all 46 of the QoL SF-36 assessments. Inside the Appendix A are data related to the total of women in the sample (64) who reported lumbar or pelvic pain.

Most of the eligible patients were in the group with ≥4 visits (89%). The mean age of pregnant women in both groups was equivalent (29, 7.1 vs. 29.9, 6.8, *p* = 0.592). The number of married/cohabitating women in the sample was the same as that of single women (50%, *p* = 0.555), while Black/Brown women were the majority (56.5%, *p* = 0.147) (Table 1).

The vast majority of the pregnant women worked outside the home (76.1%, *p* = 0.229) and earned between two and three minimum wages (62.2%); 56.5% of the sample had completed high school (*p* = 0.687). Two (4.3%, *p* = 0.131) patients lost their jobs during the study.

Table 2 shows the clinical and obstetric data of the patients. The percentage of normotensive pregnant women (69.6%, *p* = 0.655) was higher than that of women with hypertensive conditions (30.4%, *p* = 0.655). Pregnant women with gestational diabetes were more present in the group with ≤3 consultations (7/10, *p* = 0.159). Pregnant women with overweight/obesity at the final weigh-in (74.4%, *p* = 0.922) had an equivalent percentage between the two groups.

Primiparous women accounted for 30.4% (*p* = 0.277) of the sample, and the number of pregnant women with a history of previous fetal malformation was 22% (*p* = 0.575). In 24% (*p* = 0.857) of the sample, conditions that required a psychological and/or psychiatric follow-up were observed. Domestic violence cases were also verified in two (8.3%) patients in the group with ≥four OMTs. Only 13 (24%, *p* = 0.609) pregnant women were physically active.

Respiratory problems affected three (6.5%) pregnant women. COVID-19 affected 11% (*p* = 0.178) of the sample, with most cases (four) in the group with ≤3 OMTs.

Table 3 shows the results of low back and pelvic pain intensity charts among the pregnant women. Lumbar symptoms were present in 59% of the pregnant women, being prevalent (70.8%) in the group with more than three consultations. Pelvic pain was less reported in both groups by the pregnant women in the sample (28.26%). The results showed a statistically significant change in pain intensity from severe to mild in the averages of minimum and maximum values for low back pain, and moderate to mild for pelvic pain in both groups after OMT, with values falling more sharply in the group with ≥4 sessions.

The QoL SF-36 was analyzed based on its range alongside the OMTs. It is possible to verify that all the evaluated items presented a statistically significant difference in interpreting QoL scores (Table 4). The item “limitation of emotional aspects” was the one that obtained the score with the most significant variation of improvement indices (27.78 ± 40.13 vs. 55.56 ± 43.60, *p* = 0.004), and the index of general health status showed the lowest positive variation (54.04 ± 16.04 vs. 57.8 ± 14.46, *p* = 0.003).

## 4. Discussion

This study aimed to assess the intensity of lumbar and pelvic pain and the changes in QoL among pregnant women receiving OMT. The results revealed a significant improvement in both lumbar and pelvic pain intensity, as well as an enhancement in overall QoL among pregnant women who underwent OMT. The percentage of participants experiencing lumbar pain (59%) and pelvic pain (28.26%) fell within the range reported in other studies [5,6].

It is important to note that the pain intensity experienced by the pregnant women in this study did not reach a level that required hospitalization. Despite confounding elements for pain intensity, such as obesity and a sedentary lifestyle, all groups showed a statistically significant improvement in lumbar and pelvic pain. However, to gain a comprehensive understanding of the results, it is necessary to consider intrinsic subjective factors, such as the feeling of well-being, health, and satisfaction, as well as extrinsic objective factors such as living conditions and environmental influences that directly impact quality of life [15]. In addition, although this study did not specifically address COVID-19-related questions, it is essential to acknowledge the existence of a pandemic scenario that generated a health, psychosocial, and economic crisis.

The pandemic was a complex time for much of the Brazilian population, especially for at-risk groups, such as pregnant women, the elderly, and people with comorbidities. The maternity hospital where the data were collected had a maternal-fetal risk profile. Therefore, for a significant part of the sample, the risks were redoubled in the face of the COVID-19 pandemic because they were pregnant and had a maternal-fetal risk pregnancy. Undoubtedly, this risk condition was present among pregnant women in the sample. Recent research has indicated that this period significantly affected the quality of life and mental state of pregnant women in Brazil [21].

Despite the extent of COVID-19 in Brazil, all infected patients within the sample (11%) did not evolve to levels of severity requiring hospitalization, even though the pregnant women in the sample included individuals with various comorbidities, such as respiratory pathologies, obesity, hypertension, and diabetes. All patients who tested positive for COVID-19 were promptly isolated and separated from others. OMT sessions were only resumed once these individuals were asymptomatic and no longer posed a risk of spreading the disease.

Intrinsic emotional factors, such as anxiety/psychiatric illnesses (24%) and cases of domestic violence (4.3%), were observed in a significant portion of the sample. Part of the pregnant women (22%) also had a history of traumatic experiences of losing a child or having children with malformations, which left substantial and lasting traces [22] in a significant number of women. This is justified, in part, by the fact that the hospital where this study was carried out is a leading institution in the follow-up of high-risk pregnancies for the fetus.

Obesity, which affected 74% of the pregnant women in this study, is identified as the most common health problem in women of reproductive age and is accepted as an aggravating factor for joint and muscle pain, hypertension, diabetes, and depression/anxiety during pregnancy [23]. It is also recognized as a causal agent of the decrease in the QoL of pregnant women [24,25].

Equally high levels of sedentary lifestyle accompanied the expressive numbers of obesity. Although physical activity is an essential tool for controlling anxiety, obesity, the intensity of joint pain, and quality of life in the last two trimesters of pregnancy [26], it was neglected by a majority (76%) of the pregnant women.

Despite the presence of extrinsic and intrinsic factors unfavorable to the results that made up the physical component (functional capacity, physical aspects, and pain) of this study, such as the pandemic, obesity, and a sedentary lifestyle, an improvement was observed in the functional capacity of the group with ≥4 OMT and in physical aspects and pain in both groups. Significant improvements were observed even in the items related to the mental component of the quality-of-life assessment, including mental health, emotional aspects, and social aspects. It is worth noting that these improvements occurred despite the presence of patients with anxiety/psychiatric illnesses and a history of traumatic experiences within the sample. Notably, mental health and emotional aspects showed significant improvements in the group that received four or more OMT sessions.

General health status did not change in the ≤3 OMT group but increased and remained above average in the ≥4 OMT group. Despite the intense pressure on society during the pandemic, the “vitality” and “social aspects” items improved significantly in the group with ≥4 OMT.

Other recent articles [8,13,27,28] have consistently reported significant results, demonstrating the effectiveness of OMT in relieving pain among obstetric patients. These studies further support the notion that OMT can be considered a safe and beneficial complementary approach to traditional obstetric care.

Other studies on the effect of OMT on QoL and pain intensity have already demonstrated an improvement in pain among cancer patients with fibromyalgia, as well as pregnant women, using a semi-structured question survey [14,29,30]. Studies correlating the QoL associated with pain intensity of pregnant women and OMT have not yet been conducted. This study’s positive results occurred even though the research was conducted in an institution related to high maternal-fetal risk pregnancies and during a severe health, psychosocial, and economic crisis caused by COVID-19.

However, data captured during the pandemic can be seen both in favor of this study and also as a limitation, a bias. It can be regarded as favorable because it allows for an evaluation of the influence of OMT in a group of women doubly exposed to risks, both to those related to the pandemic and those related to a pregnancy with maternal-fetal risk. The stress to which these pregnant women were submitted is undoubtedly more significant than that experienced by the average woman who was not exposed to many adverse situations. This study thus made it possible to assess how much OMT improved quality of life and decreased the intensity of lumbar and pelvic pain in most of the pregnant women despite having considerable losses in the sample.

On the other hand, the pandemic can be considered a bias with regards to the emotional state of a part of the sample. The mental state resulting from the socioeconomic-sanitary situation resulting from COVID-19 may have altered the habits and behaviors of the participants. Studies on the effects of the pandemic on pregnant women also report a reduction in physical activity, an increase in negative eating habits [31], and an increase in domestic violence [32]. In a biopsychosocial model, these factors were identified as determinants for stress during pregnancy [33] that may have contributed to the sample’s psychological suffering and pain condition.

The improvement in all quality-of-life and pain indices observed in the sample submitted to OMT is certainly also correlated with the ability of therapeutic touch to wake up interoceptive, proprioceptive, and exteroceptive receptors, generating a cross-modal network capable of altering musculoskeletal tensions [34]. This communication allows for a correspondence between our internal and external environment, connecting the perception of inner well-being to what we perceive externally [35]. Neurophysiological effects added to contextual effects, such as the nocebo or placebo effect, answer why manual practices can alter the patient’s perception of pain [36], consequently altering their quality of life.

According to one review, manual therapy may influence interoceptive deficits associated with various physical and psychological health conditions. Interoception is a complex mind–body experience, and proprioceptive or exteroceptive touch and interoceptive networks can transmit emotionally valuable information to the insula [37]. Thus, “the sensations processed by discriminative and affective pathways activate different mechanisms in the somatosensory and insular cortex, respectively” [37]. Good OMT responses are also owed to the union between touch, manipulation, and mobilization that triggers neurophysiological responses, improving movement, decreasing pain, changing the biomechanics of elasticity, and increasing tissue flexibility [38,39].

Despite the good data observed, this study had other limitations that must be considered when evaluating the results. One limitation was the need to exclude data from pregnant women who had only completed the QoL SF-36 once. This loss may be related to the way the data collection was performed, i.e., delivering the SF-36 questionnaire to pregnant women upon arrival at the service and collecting this instrument at the end. This was because these pregnant women underwent multiple assessments during the consultation. It was considered that the most appropriate procedure would be to let them fill out the questionnaire when they felt more comfortable doing so. Hence, many pregnant women chose not to complete the questionnaire or to fill it out partially, thus reducing the number of eligible patients.

The SF-36 itself can also be considered a limitation since it does not make any corrections according to the gestational age of the pregnancy. This correction is necessary since the reasons for pain may be related only to hormonal changes or uterus enlargement, being physiologically expected. For this reason, we chose to work with the SF-36 score variation instead of the first and last evaluation. The hypothesis that OMT caused changes in SF-36 scores overtime was worked on, and the results of this study demonstrate that pregnant women with more OMT sessions tended to show more significant improvements in SF-36 scores.

The clinical relevance of this study is because QoL questionnaires are tools that give the patient a voice to express their physical, psychological, and emotional impressions about the therapy used, improving the quality of healthcare models. In particular, the QoL SF-36 is a questionnaire that stands out for verifying the effectiveness of health interventions [16]. The positive results of statistically significant improvements in all eight items that make up the questionnaire validate OMT as an effective therapeutic tool in improving the QoL of pregnant women in the third trimester of pregnancy.

## 5. Conclusions

Despite the presence of extrinsic and intrinsic confounding factors within the patients and the small sample size, the results of this study robustly demonstrate the effectiveness of OMT in reducing pain intensity and improving the quality of life (QoL) of pregnant women. The findings highlight that OMT is an effective practice for reducing the intensity of low back and pelvic pain, thus constituting an improvement in the QoL of pregnant women in the third trimester of pregnancy.

However, further studies with more robust samples will be needed to confirm the strength of the data observed.

## Figures and Tables

**Figure 1 healthcare-11-02538-f001:**
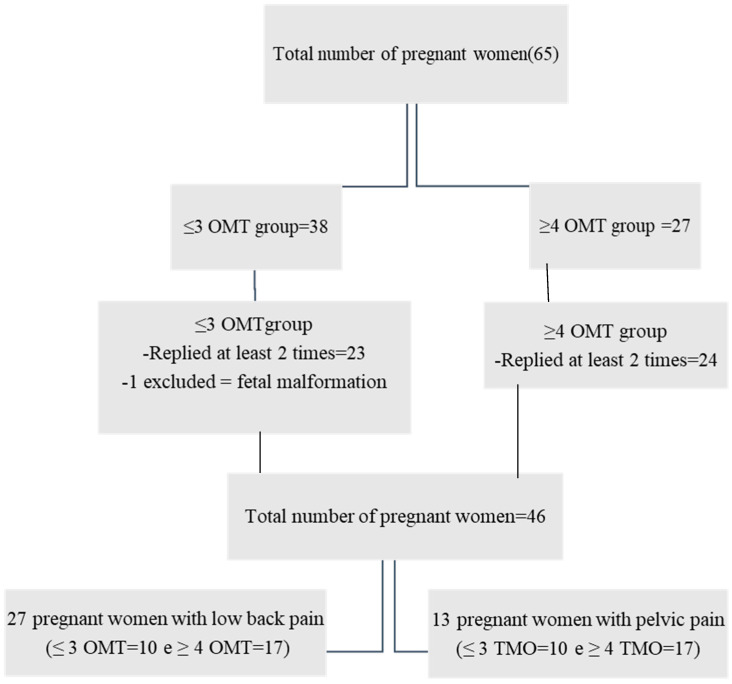
Flowchart with the total number of pregnant women who entered the study and the total number of women who completed the QoL SF-36 at least twice.

**Table 1 healthcare-11-02538-t001:** Demographic data of the sample.

Variable	≤3 OMT’s ^1^, N = 22	≥4 OMT’s, N = 24	Total	*p*-Value ^2^
**Age**				
Mean, SD	29.0, 7.1	29.9, 6.8		0.592
Min, max	19.0, 44.0	20.0, 44.0		0.592
Unknown	1	0		
**Marital status**				
Married/cohabitating	10 (45.5%)	13 (54.2%)	23 (50%)	0.555
Single ^3^	12 (54.5%)	11 (45.8%)	23 (50%)	0.555
**Color/race**				
White	12 (54.5%)	8 (33%)	20 (43.5%)	0.147
Black/Brown	10 (45.5%)	16 (66.7%)	26 (56.5%)	0.147
**Family income**				
<2 minimum wages	6 (28.6%)	3 (12.5%)	9 (20%)	0.391
2 to 3 minimum wages	12 (57.1 %)	16 (66.7%)	28 (62.2%)	0.391
≥3 minimum wages	3 (14.3%)	5 (20.8%)	8 (17.8%)	0.391
Unknown	1 (4.54%)			0.391
Employed	15 (68.2%)	20 (83.3%)	35 (76.1%)	0.229
Job loss	2 (9.1%)	0 (0.0%)	2 (4.3%)	0.131
**Education**				
Elementary School	4 (18.2%)	3 (12.5%)	7 (15.2%)	0.687
High school	11 (50%)	15 (62.5%)	26 (56.5%)	0.687
University education	7 (31.8%)	6 (25%)	13 (28.3%)	0.687

^1^ abbreviation: OMT = osteopathic manipulative treatment; ^2^ Pearson’s chi-squared test; ^3^ Being single does not mean not having a partner. Only one pregnant woman in this sample did not have a partner.

**Table 2 healthcare-11-02538-t002:** Clinical and obstetric data of the sample.

Variables	≤3 OMT’s *, N = 22	≥4 OMT’s, N = 24	Total	*p*-Value ^1^
**Normotensive**	16 (73%)	16 (67%)	32 (69.6%)	0.655
**Hypertensive Conditions ^2^**	6 (27.3%)	8 (33.3%)	14 (30.4%)	0.655
**Diabetes**	7 (31.8%)	3 (12.5%)	10 (22%)	0.159 ^2^
**Weight rating ^3^**				
**Initial weight obesity risk ^4^**				
Eutrophic	7 (35%)	7 (29.2%)	14 (32%)	0.679
Obese/overweight	13 (65%)	17 (71%)	30 (68.2%)	0.679
**Final weight obesity risk**				
Eutrophic	5 (26.3%)	6 (25%)	11 (25.6 %)	0.922
Obese/overweight	14 (74%)	18 (75%)	32 (74.4%)	0.922
Unknown	1 (4.54%)	0 (100%)	1 (2.17)	0.922
**Primiparous**	5 (28%)	9 (37.5%)	14 (30.4%)	0.277
**History of fetal malformation**	4 (18.2%)	6 (25%)	10 (22%)	0.575
**Anxiety/psychiatric condition**	5 (22.7%)	6 (25%)	11 (24%)	0.857
**History of domestic abuse/violence**	0 (0%)	2 (8.3%)	2 (4.3%)	0.166
**Physical activity**	6 (27.3%)	5 (20.8%)	11 (24%)	0.609
**Lung disease ^5^/asthma**	1 (4.5%)	2 (8.3%)	3 (6.5%)	0.95
**COVID**	4 (18.2%)	1 (4.2%)	5 (11%)	0.178 ^4^
N (%)				

Abbreviations: OMT * = osteopathic manipulative treatment. ^1^ Pearson’s chi-squared test, Fisher’s exact test; ^2^ Gestational hypertension = group composed of pregnant women with gestational hypertension (5), chronic hypertension (08), and PE (02); ^3^ BMI based on the Institute of Medicine (IOM 2009)—relative data and final obesity; ^4^ initial weight is pre-gestational and reported; ^5^ There were two cases of asthma, one in each group. The lung disease was sleep apnea.

**Table 3 healthcare-11-02538-t003:** Minimum and maximum pain scores reported by participants before and after OMT.

Pain Score	Group ≤ 3			Group ≥ 4		
	Before	After	*p*-Value	Before	After	*p*-Value
**Low back pain score min**	6.5 ± 1.8	4.0 ± 1.8	0.007	5.2 ± 2.1	2.6 ± 1.4	0.001
**Low back pain score max**	7.4 ± 1.6	4.1 ± 1.7	0.005	7.6 ± 1.5	3.6 ± 1.8	<0.001
**Pelvic pain score min**	7.2 ± 1.7	3.7 ± 2.0	0.027	4.3 ± 1.7	1.7 ± 0.5	0.017
**Pelvic pain score max**	7.2 ± 1.7	3.7 ± 2.0	0.027	6.0 ± 2.6	2.0 ± 0.8	0.018

**Table 4 healthcare-11-02538-t004:** Range of quality of life scores reported by participants in relation to the number of OMT sessions performed before delivery.

	Group ≤ 3			Group ≥ 4		
	Score Min	Score Max	*p*-Value	Score Min	Score Max	*p*-Value
**Functional capacity**	62.5 ± 28.5	62.9 ± 28.3	0.317	46.0 ± 22.5	61.5 ± 22.0	0.001
**Physical aspect limitation**	28.6 ± 37.8	30.4 ± 36.9	0.318	14.6 ± 26.5	30.2 ± 33.8	0.004
**Pain**	36.3 ± 17.6	37.2 ± 16.4	0.319	34.9 ± 13.5	47.3 ± 19.0	0.001
**General state**	47.8 ± 23.0	47.8 ± 23.0	1.000	54.0 ± 16.0	57.8 ± 14.5	0.003
**Vitality**	35.8 ± 17.5	35.8 ± 17.5	1.000	27.5 ± 16.2	39.0 ± 17.5	0.001
**Social aspects**	57.7 ± 30.4	58.7 ± 29.9	0.317	53.1 ± 26.4	66.1 ± 24.9	0.002
**Emotional aspects limitations**	43.6 ± 39.4	43.6 ± 39.4	1.000	27.8 ± 40.1	55.6 ± 43.6	0.004
**Mental health**	55.1 ± 23.2	55.1 ± 23.2	1.000	53.3 ± 22.0	61.5 ± 21.9	0.001

## Data Availability

All data are available in https://doi.org/10.35078/SHDD3G.

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
