# Peer review of "Influence of Osteopathic Manipulative Treatment on the Quality of Life and the Intensity of Lumbopelvic Pain in Pregnant Women in the Third Trimester: A Prospective Observational Study"

_healthcare, 2023, doi:10.3390/healthcare11182538_

Round 1

Reviewer 1 Report

Thank you for the opportunity to review this manuscript. I have made many notes on the file, which is attached here for your reference. I have two primary concerns with this manuscript, which must be addressed before it can proceed. 

1) Please explain and justify the division of the sample into two groups. This appears to be an arbitrary post hoc attempt to make a comparison implying that more osteopathic consultations leads to better outcomes, however, the sample may be more robust when viewed as a single cohort and results analysed for changes within the group over time. 

2) Please explain why non-parametric analyses were use for the continuous variables? Were the data skewed or non-normally distributed? 

Generally, the English language expression is sufficient. 

Author Response

Responses to reviewer 1

1)- Please explain and justify the division of the sample into two groups. This appears to be an arbitrary post hoc attempt to make a comparison implying that more osteopathic consultations leads to better outcomes, however, the sample may be more robust when viewed as a single cohort and results analysed for changes within the group over time.

Response: In fact, the reviewer is correct. The division of the sample was decided after the initial analyses when we verified that the group was stratified into two groups (up to 3 sections and greater than 3). Our objective was not to verify how much the number of sections impacted on the quality of life. Our objective was to verify which participant profile characterized the groups. We have the analysis considering the number of sections, and we can include it if you think it is necessary.

The division of the groups also occurred by the follow-up time each participant had at the study. The maternity where the data were collected has a fetal risk profile. Thus, pregnant were referred according to their risk and the need specialized follow-up. The stratify by follow-up time also helped to homogenize the participants.

We change and put it in the text: The division of the sample was decided after the initial analyses when we verified that the group was stratified into two groups. The evaluated groups were defined considering the number of OMT performed in ≤ 3 and ≥4 sections.

2)- Please explain why non-parametric analyses were use for the continuous variables? Were the data skewed or non-normally distributed? 

Response: We performed non-parametric tests because the data were not a normal distribution.

We checked the normal distribution of the data using the Kolmogorov-Smirnov test and all the p-values were less than 0.05, rejecting the hypothesis of normal distribution. We don't know the reason for the observed asymmetry. However, we believe that it may be related to the socio-economic profile of the participants. The hospital where the data was collected receives high-risk fetal patients from different social levels (rich and poor). The perception of quality of life has an important socio-economic component, which could explain the variation between responses. We also believe that part of the variation could be related to variations in the gestational age of each patient. Unfortunately, the small sample size did not allow us to stratify the population by gestational age.

Reviewer 2 Report

algorithm on p. 4...last box on right... written as TMO = 10 and TMO+17  I think error

Author Response

Reviewer 2

algorithm on p. 4...last box on right... written as TMO = 10 and TMO+17  I think error

Answer: Yes, it was a mistake. Corrected

Reviewer 3 Report

Thank you for the opportunity to read the research.

In my opinion, the authors fail to justify the choice of the topic and the purpose of the research. There is no introduction to physiotherapy in pregnant women.

Line 60: COVID is irrelevant here (similar impression in 204 - 209). Why were pregnancies 27-41 included in the study?

Primiparous - should be considered separately.

Activity? - to expand topic.

Were the patients' nutrition monitored?

Two scales is not enough in my opinion? Why weren't other measurements used?

No figure caption on page 4.

Table 1,2,4 to be reorganized, bold and color make the table hard to read.

Table 3 illegible - poor quality.

Comparing studies to studies by other authors is insufficient - further explanation is needed even if the number of studies is small.

Another font of footnotes.

Author Response

Reviewer 3

1-In my opinion, the authors fail to justify the choice of the topic and the purpose of the research. There is no introduction to physiotherapy in pregnant women.

Answer-I want to clarify that this article concerns osteopathic treatment, not physiotherapy. Although both professions use the hands to treat, osteopathy is not part of physiotherapy. Especially because osteopathy precedes physiotherapy in time, it was founded in 1874.
According to the WHO, osteopathy is an independent therapy with its philosophy, theoretical basis, training, and practices, which can be verified in the Benchmarks for Training in Osteopathy, launched by the WHO in 2010 (ISBN 9789241599665).

2-Line 60: COVID is irrelevant here (similar impression in 204 - 209). Why were pregnancies 27-41 included in the study?

Answer- I think COVID is not irrelevant. Brazil was one of the countries with the most deaths in the world. These pregnant women were doubly at risk due to the dangers of covid during pregnancy and because they were already women at high maternal-fetal risk. Vaccination took a while to arrive in Brazil. Pregnant women were advised to avoid going out, working from home, and, like everyone else, avoiding crowds. It was a stressful period. This affected people's quality of life and limited physical activities. When researching a life questionnaire, we cannot forget the extrinsic factors. If we research pain, how can we forget that people were advised not to leave their homes?

About 27-41, it was a mistake. The correct sentence is: All pregnant women from 27 of gestation over 18 years of age who were followed up by the prenatal service of the institution were considered eligible.

3-Primiparous - should be considered separately.

4-Activity? - to expand topic.

Answer- Are you talking about physical activity? Do you want to say what kind of physical activity they did?

I will explain better.

5-Were the patients' nutrition monitored?

Answer- Nutrition or BMI? We observed the BMI because obesity is an essential factor in increasing pain, and according to a study, it could promote depression.

6-Two scales is not enough in my opinion? Why weren't other measurements used?

Answer- Could you tell me from which place in the text you are saying?

7-No figure caption on page 4.

Answer- In fact, the flowchart was there, but the title of Figure 1 was absent. It will be corrected.

8-Table 1,2,4 to be reorganized, bold and color make the table hard to read.

Answer- it will be corrected.

Table 3 illegible - poor quality.

Answer- it will be corrected.

9-Comparing studies to studies by other authors is insufficient - further explanation is needed even if the number of studies is small.

Answer- Your statement is not clear to me. I think you want me to put the data of the studies and not just comment on their existence. If so, I can do it.

10-Another font of footnotes.

Answer- I didn´t understand.

Reviewer 4 Report

I read with interest your manuscript about the effects of osteopathic manipulative treatment in the quality of life of pregnant women in the third semester. The first questions that arises is why you focused in quality of life, although also collected pain data. This seems a bit poor regarding other studies published. Moreover, after reading the manuscript several times, it is difficult for me to decided if this is an observational or experimental study, point that must be cleared in the title and Study Design section. Nevertheless, the study could be highly improved if the authors make an effort. Please consider my suggestions: 

Title: please include the type of study

Introduction:

Line 12: a space is missing before “Both”

Line 22: please indicate the instrument use to assess pain and quality of life in order to facilitate the comprehension of the results.

Lines 44-47: have not those studies considered the quality of life of pregnant women? What new information offers your study to the existent knowledge?

Line 52: please cite that “few data”

Materials and Methods:

Line 58: is this study an experimental or an observational one? Describe the design better.

Line 73-79: surprisingly, authors decided to change the reference style in these lines to APA, using authors names and year of publication. Please check the correct style, as it seems a mistake of adaptation from the style of another journal.

Line 89: did you mean “QoL SF-36”?

Line 93: at this section, the study seems to be an experimental one, not observational. In this line, authors must state why trial registry is not required.

Line 107: once again a wrong citation style.

Line 108: is the Brazilian version of the questionnaire validated? Please indicate its psychometric properties.

Line 109: please include the expected time a patient needs to fulfill the questionnaire.

Line 125 and 126: your repeated “categorical variables” twice while describing different analysis. Please correct.

Line ??: I see nowhere the group allocation described in the abstract. This added to the lack of a trial registry make me think that authors have messed up the designs and data…

Line 134-135: here I found the group allocation. This is not a result, this must be in methods section. Furthermore, why are participants divided according the number of visits?

Figure 1: it has no title and is full of grammatical mistakes as the lack of capital letters for beginning the sentences or some spaces missing. Please make a new one.

Table 1, 2, 3 and 4: they don’t follow the journal rules. 

Table 3: poor quality, it is difficult to read.

Lines 192-196: authors point to several difficulties due to pandemic scenario, but this is the first time the reveal something about when was the study performed. Maybe they should include the dates in methods section.

Line 204-206: a contingency plan would be valuable in the methods section considering a possible Covid-19 positive among participants.

DISCUSSION

This section must relate authors findings with the ones of other authors. In this concern, I suggest the authors to organize the Discussion in less paragraphs, as they use 17 different (up to eight of them with four lines or less) for talk about the two variables measured and the concerns about pandemic scenario. Please join those paragraphs about the same variables including your findings and comparisons with other authors. By these, the section will look clearer and will be easier to be followed by any reader.

The English looks fine for me, nonetheless I recommend to be checked by a native speaker reviewer.

Author Response

Reviwer4

I read with interest your manuscript about the effects of osteopathic manipulative treatment in the quality of life of pregnant women in the third semester. The first questions that arises is why you focused in quality of life, although also collected pain data. This seems a bit poor regarding other studies published. Moreover, after reading the manuscript several times, it is difficult for me to decided if this is an observational or experimental study, point that must be cleared in the title and Study Design section. Nevertheless, the study could be highly improved if the authors make an effort. Please consider my suggestions: 

1-Title: please include the type of study

Answer: done

2-Introduction:

A-Line 12: a space is missing before “Both”

Answer: adjusted

B-Line 22: please indicate the instrument use to assess pain and quality of life in order to facilitate the comprehension of the results.

Answer: QoL SF-36 for QoL and visual pain scale. Both are in supplementary materials.

C-Lines 44-47: have not those studies considered the quality of life of pregnant women? What new information offers your study to the existent knowledge?

Answer: First, this study was conducted during a pandemic period. It was a stressful period, especially in Brazil and for pregnant women. Pain and QoL are sensitive to this. In addition, to date, no study has been conducted relating the influence of OMT on QoL in high-risk pregnancies.

D-Line 52: please cite that “few data”

Answer: Few studies 

3-Materials and Methods:

E-Line 58: is this study an experimental or an observational one? Describe the design better.

Answer: This is an observational study. If it was a Clinical trial, we had to randomize the sample, make a sample calculation, and divide it into group control and intervention, for example. All the groups are exposed to OMT. Our study is an observation of the influence of OMT over lumbopelvic pain and QoLin pregnant women. Our research observes and analyzes events after OMT.

F-Line 73-79: surprisingly, authors decided to change the reference style in these lines to APA, using authors names and year of publication. Please check the correct style, as it seems a mistake of adaptation from the style of another journal.

Answer:   adjusted

G-Line 89: did you mean “QoL SF-36”?

Answer:  Yes. It was written (To enable enough time for answering the QoL SF-36, the patient's entry into the study had a deadline of 36 weeks of gestational age (GA).)

H-Line 93: at this section, the study seems to be an experimental one, not observational. In this line, authors must state why trial registry is not required.

Answer:  In this case, the exposure is OMT. In observational studies, the group is exposed to something, including a treatment or medication. 

I-Line 107: once again a wrong citation style.

Answer:  adjusted 

J-Line 108: is the Brazilian version of the questionnaire validated? Please indicate its psychometric properties.

Answer:  Yes, it is validated. It was adapted according to simple Portuguese, matching our cultural values. It was compared with the German, French, and Swedish versions, with cultural adaptations such as measurements from miles to kilometers and assessment activities such as functional ability. Inter and intra-observer reproducibility was observed.

For more information you can read at :

  1. Ciconelli RM, Ferraz MB, Santos W, Meinão I, Quaresma MR. Tradução para a língua portuguesa e validação do questionário genérico de avaliação de qualidade de vida SF-36 (Brasil SF-36). Rev Bras Reumatol. junho de 1999;39:143–50.
  2. Campolina AG, Ciconelli RM. O SF-36 e o desenvolvimento de Novas Medidas de Avaliação de Qualidade de Vida. Acta Reumatol Port. 2008;33:127–33.

K-Line 109: please include the expected time a patient needs to fulfill the questionnaire.

Answer:  It was not a precise time. They used the time they needed.

L-Line 125 and 126: your repeated “categorical variables” twice while describing different analysis. Please correct.

Answer:  adjusted

M-Line ??: I see nowhere the group allocation described in the abstract. This added to the lack of a trial registry make me think that authors have messed up the designs and data…

Answer:  This is an observational study, not a randomized clinical trial

N-Line 134-135: here I found the group allocation. This is not a result, this must be in methods section. Furthermore, why are participants divided according the number of visits?

Answer: It was there in lines 124-125.

The division of the sample was decided after the initial analyses when we verified that the group was stratified into two groups (up to 3 sections and greater than 3). Our objective was to verify which participant profile characterized the groups. The division of the groups also occurred by the follow-up time each participant had at the study. The maternity where the data were collected has a fetal risk profile. Thus, pregnant were referred according to their risk and the need for specialized follow-up. The stratify by follow-up time also helped to homogenize the participants. 

O-Figure 1: it has no title and is full of grammatical mistakes as the lack of capital letters for beginning the sentences or some spaces missing. Please make a new one.

 Answer: done

P-Table 1, 2, 3 and 4: they don’t follow the journal rules.

 Answer: done

Q-Table 3: poor quality, it is difficult to read.

 Answer: done

R-Lines 192-196: authors point to several difficulties due to pandemic scenario, but this is the first time the reveal something about when was the study performed. Maybe they should include the dates in methods section.

 Answer: This is already there : This prospective observational study assesses the influence of OMT on QoL and the intensity of low back and pelvic pain in pregnant women in the third trimester of pregnancy. The survey was carried out between August 2021 and September 2022 during the COVID-19 pandemic in Brazil. (lines 62-63).

S-Line 204-206: a contingency plan would be valuable in the methods section considering a possible Covid-19 positive among participants.

Answer:  All patients with symptoms of covid were tested. They were not allowed to enter the hospital, and when they were diagnosed inside the hospital, they were isolated. None of the pregnant women in the sample who contracted covid were admitted to the hospital, nor were they treated with OMT during contamination by the covid virus.

4-DISCUSSION

This section must relate authors findings with the ones of other authors. In this concern, I suggest the authors to organize the Discussion in less paragraphs, as they use 17 different (up to eight of them with four lines or less) for talk about the two variables measured and the concerns about pandemic scenario. Please join those paragraphs about the same variables including your findings and comparisons with other authors. By these, the section will look clearer and will be easier to be followed by any reader.

Answer: adjusted

Reviewer 5 Report

Dear authors,

This study is interesting, and I think it is relevant to the scientific community. However, I have noted some areas of improvement as follows:

  • My first note is about the reference style the authors used. In the introduction, they used the Vancouver style, but in the methods, they changed the style to the APA, and then they switched back to Vancouver in the discussion. I suggest considering only one citation style and using it consistently throughout the manuscript.
  • The introduction could be improved to include a more in-depth analysis of the current body of literature on the topic.
  • The methods section requires significant revision. For example, there is no information about the sampling method and the sample size calculations. I suggest following the STROBE framework in reporting this section. The way the methods are presented here is misleading and lacks important details about the approaches the authors considered in collecting their data. It is hard to assess the study’s quality and risk of bias based on the information the authors provided.
  • In lines 125-126, the authors cited, “The Chi-square test was used to identify statistically significant differences in categorical variables and the Mann-Whitney test for categorical variables.” Please revise.
  • The results section seems ok with a recommendation to improve the presentation of figures and tables if possible.
  • The discussion requires revision. To me, at times, the authors are not discussing their findings; instead, they are presenting some unnecessary arguments. For instance, they discussed the intrinsic and extrinsic factors that may influence their pain and quality of life; however, the study's main objective is to examine the role of OMT in relieving pain and QoL of pregnant women. I think all the psychosocial factors mentioned in the discussion can be presented in the introduction section as background information that helps contextualize their study.
  • I think the authors should also include the timing of their study during the COVID-19 pandemic as a limitation. Also, they should mention the small sample size and its nuances in their conclusion.
  • Finally, some language typos are noted. I suggest proofreading the entire manuscript.

Otherwise, I think the topic of this manuscript is interesting and valuable.

All the best,

Several typos were detected. I suggest proofreading. 

Author Response

Reviewer 5

This study is interesting, and I think it is relevant to the scientific community. However, I have noted some areas of improvement as follows:

-My first note is about the reference style the authors used. In the introduction, they used the Vancouver style, but in the methods, they changed the style to the APA, and then they switched back to Vancouver in the discussion. I suggest considering only one citation style and using it consistently throughout the manuscript.

  • Answer: It was adjusted

-The introduction could be improved to include a more in-depth analysis of the current body of literature on the topic.

-The methods section requires significant revision. For example, there is no information about the sampling method and the sample size calculations. I suggest following the STROBE framework in reporting this section. The way the methods are presented here is misleading and lacks important details about the approaches the authors considered in collecting their data. It is hard to assess the study’s quality and risk of bias based on the information the authors provided.

  • Answer: It was adjusted

-In lines 125-126, the authors cited, “The Chi-square test was used to identify statistically significant differences in categorical variables and the Mann-Whitney test for categorical variables.” Please revise.

  • Answer: It was adjusted
  • The results section seems ok with a recommendation to improve the presentation of figures and tables if possible.
  • Answer: done

-The discussion requires revision. To me, at times, the authors are not discussing their findings; instead, they are presenting some unnecessary arguments. For instance, they discussed the intrinsic and extrinsic factors that may influence their pain and quality of life; however, the study's main objective is to examine the role of OMT in relieving pain and QoL of pregnant women. I think all the psychosocial factors mentioned in the discussion can be presented in the introduction section as background information that helps contextualize their study.

  • Answer: Other researchers raise the discussion about the importance of intrinsic and extrinsic factors in QoL. Reference to this appears in the manuscript. Removing social psychic factors from an analysis of quality of life, in my point of view, and from the author used as a reference, would be a mistake. Both types of pain has emotional components, as well as quality of life. Stress and a pandemic directly affect the analyzed aspects. That´s why we also discuss it in discussion.

-I think the authors should also include the timing of their study during the COVID-19 pandemic as a limitation. Also, they should mention the small sample size and its nuances in their conclusion.

  • Answer: I agree.

-Finally, some language typos are noted. I suggest proofreading the entire manuscript.

  • Answer: It was adjusted

Round 2

Reviewer 3 Report

Thank you for the corrections.

1. Line 66-67: I still believe that there is no need to emphasize the pandemic

2. Primiparous - should be considered separately

3. not BMI, I asked about keeping food diaries and meal control

4. Why were only QoL SF-36 and VAS used?

5. the tables still need to be unified

6. you need to change the font in the section: references

Author Response

Reviewer 3

  1. Line 66-67: I still believe that there is no need to emphasize the pandemic.

Answer: I'm sorry, but there is a conflict here. One of the reviewers found it necessary, and I agree. The moment of the pandemic in Brazil was terrible and intensely affected people's emotions and lives, especially pregnant women who were at risk. It is a way of contextualizing a situation. It was not a period of normality in people's lives.

  1. Primiparous - should be considered separately.

Answer: In table 2, there were put. In line 169 also. But there were in our sample nothing more different to show about them.

  1. not BMI, I asked about keeping food diaries and meal control.

Answer: I´m sorry, but for the OMT, this is not relevant. However, BMI is important because body mass greatly influences pain and quality of life.

  1. Why were only QoL SF-36 and VAS used?

Answer: Because with these two tests, we can observe both the evolution of pain and changes in the patient's quality of life. It was enough within what we would like to get data. This article is part of a more extensive study in which other aspects and evaluations were observed.

  1. the tables still need to be unified.

Answer: All tables were made in Excel. They were corrected and unified in the first round of revision.

  1. you need to change the font in the section: references.

 Answer: It was done.

Reviewer 4 Report

I want to thank the authors for all the changes performed. Despite this, I think that some of my comments were not fully understandable as some changes are still missing:

-Line 108: thank you of the references about the validation of the questionnaire. Please include them as references in the text.

-LINES 204-206: I fully understand the difficulties we all experiences during the pandemic situation in personal life and work too. Considering that the pandemic situation could be both a weak point and a strength regarding this study, the point that could be considered a strength is if you explain your specific contingency plan in case a covid-19 positive appear among participants. Please dedicate a paragraph of methods to explain everything you did in this line: Covid tests you performed to participants, if masks were used, if assessments were performed in ventilated rooms, if rooms were cleaned after each participant assessment... anything could be a strength point for the methods of your study.

S-Line 204-206: a contingency plan would be valuable in the methods section considering a possible Covid-19 positive among participants.

Author Response

Reviewer 4

-Line 108: thank you of the references about the validation of the questionnaire. Please include them as references in the text.

Answer: Done

Lines 121-125-The Brazilian version of the QoL SF-36 was adapted according to simple Portuguese, matching our cultural values. It was compared with the German, French, and Swedish versions, with cultural adaptations such as measurements from miles to kilometers and assessment activities such as functional ability. Inter and intra-observer reproducibility was observed.

-LINES 204-206: I fully understand the difficulties we all experiences during the pandemic situation in personal life and work too. Considering that the pandemic situation could be both a weak point and a strength regarding this study, the point that could be considered a strength is if you explain your specific contingency plan in case a covid-19 positive appear among participants. Please dedicate a paragraph of methods to explain everything you did in this line: Covid tests you performed to participants, if masks were used, if assessments were performed in ventilated rooms, if rooms were cleaned after each participant assessment... anything could be a strength point for the methods of your study.

Answer: Done.

lines 97-101 2.3.1 Pandemic procedures

As a result of the period of care occurring during the pandemic, the use of masks in hospital facilities has become mandatory for staff and patients. When vaccines began to be made available through the public health system, proof of vaccination was also required to enter the hospital.

Lines110-113- The consultations at the OMT took place following the hospital protocol, which recommended keeping the windows open to improve the ventilation of the environment and minimize the possibility of contagion. Patients were treated dressed and placed on disposable sheets.

S-Line 204-206: a contingency plan would be valuable in the methods section considering a possible Covid-19 positive among participants.

Answer: In lines 233-237- There is already some information about patients with suspected or diagnostic COVID-19.: All patients who tested positive for COVID-19 were promptly isolated and separated from others. OMT sessions were only resumed once these individuals were asymptomatic and no longer posed a risk of spreading the disease.